# Antibacterial Activities of Monsonia Angustifolia and Momordica Balsamina Linn Extracts against Carbapenem-Resistant Acinetobacter Baumannii

**DOI:** 10.3390/plants11182374

**Published:** 2022-09-12

**Authors:** Noel-David Nogbou, Dimpho Raesibe Mabela, Buang Matseke, Ntwanano Sipho Mapfumari, Mmammosheledi Elsie Mothibe, Lawrence Chikwelu Obi, Andrew Munyalo Musyoki

**Affiliations:** 1Department of Microbiological Pathology, School of Medicine, Sefako Makgatho Health Sciences University, P.O. Box 211, Pretoria 0208, South Africa; 2Department of Pharmaceutical Sciences, School of Pharmacy, Sefako Makgatho Health Sciences University, P.O. Box 218, Pretoria 0208, South Africa; 3Department of Pharmacology, School of Pharmacy, Rhodes University, P.O. Box 94, Grahamstown 6139, South Africa; 4Faculty of Science and Technology, Sefako Makgatho Health Sciences University, P.O Box 138, Pretoria 0204, South Africa

**Keywords:** antibacterial activity, plant extract, *M. balsamina Linn*, *M. angustifolia*, carbapenemase-producing *A. baumannii*

## Abstract

Carbapenemase-producing *Acinetobacter baumannii* (*A. baumannii*) is resistant to most of the available antibiotics and poses serious therapeutic challenges. The study investigated *Monsonia angustifolia* (*M. angustifolia*) and *Momordica balsamina Linn* (*M. balsamina Linn*) extracts for antibacterial activity against a clinical isolate of carbapenemase-producing *A. baumannii* using the Kirby Bauer disc diffusion and TLC coupled with bioautography. MIC determination experiments were conducted on a molecularly characterized *A. baumannii* isolate identified using VITEK2. Positive PCR detection of *bla_OXA-51_* and *bla_OXA-23_* confirmed isolate identity and the presence of a carbapenemase-encoding gene. Antibacterial activity was observed with the methanolic extract of *M. balsamina Linn* with a MIC of 0.5 mg/mL. Compounds with Rf values of 0.05; 0.17; 0.39 obtained from *M. angustifolia* hexane extract; compounds with Rf values of 0.58; 0.78; 0.36; 0.48; 0.5; 0.56; 0.67; 0.9 obtained from *M. angustifolia* dichloromethane extract; compounds with Rf values of 0.11; 0.56; 0.24; 0.37 obtained from *M. angustifolia* acetone extract and compounds with Rf values of 0.11; 0.27 obtained from *M. angustifolia* methanol extract demonstrated a level of antibacterial activity. *M. angustifolia* and *M. balsamina Linn* plant extracts have a clinically significant antibacterial activity against a carbapenemase-producing *A. baumannii* strain.

## 1. Introduction

The emergence and rapid spread of multidrug-resistant microorganisms in clinical settings has reportedly complicated empirical therapeutic management of patients [1,2] The therapeutic management of infectious diseases mainly relies on the efficacy of antimicrobial drugs [3]. Unfortunately, misuse, overuse, lack of appropriate antibiotic stewardship and adequate policy addressing the use of antibiotics have contributed to decreased efficacy of antimicrobial drugs [4]. Globally, antimicrobial resistance is of public health concern [5]. Prolonged length of hospitalization and increased hospital mortality have been identified as the direct outcome of multidrug-resistant infection [6]. *Acinetobacter baumannii* (*A. baumannii*) is a Gram-negative, obligate aerobe, coccobacillus [7,8] bacterium identified by the World Health Organization (WHO) as one of the top critical priority pathogens for which new therapeutic alternatives are urgently needed [9]. The pathogen is responsible for a range of infections including ventilator-associated pneumonia [7], central line-associated bloodstream infections [10], catheter-associated urinary tract infections [11] and may occasionally cause skin and soft tissue infection at surgical sites [12] often seen in patients admitted to intensive care units. Strains of *A. baumannii* have developed and/or acquired several resistance mechanisms to resist antibiotic action and ensure bacterial survival in antibiotic attacks [13]. Carbapenems are among the most efficient antibiotics with a broad-spectrum antibacterial activity [14]. Resistance to carbapenem drugs is challenging for clinicians as the drugs are used for management of acute Gram-negative bacterial infections [15]. Carbapenemase-producing strains of *A. baumannii* become resistant to carbapenems through enzymatic degradation of carbapenems by beta-lactamase enzyme production following acquisition and/or intrinsic expression of carbapenem resistance-associated genes [16]. Resistance to beta-lactams, macrolides, aminoglycosides, tetracyclines, fluoroquinolones, lincosamides, streptogramin and polymyxin classes of antibiotics has been documented [13]. There is a global public health concern that *A. baumannii* infections will soon be untreatable [17]. Multilateral organizations and international institutions have called for the development of alternative therapeutic management and/or new drugs to address this growing threat to human health [9,18,19].

Plants are known to demonstrate significant pharmacological properties such as antioxidant [20], antimicrobial [21,22], anti-inflammatory [23] and anticancer [24] properties that are attributed to their phytochemical compounds. Several herbal remedies have been used by indigenous medicine practitioners to treat infectious diseases due to Gram-negative bacteria such as typhoid fever due to *Salmonella typhimurium* and diarrhea due to *Escherichia coli* [25]. Plants offer an array of antimicrobial potential with reported activities against some of the top priority pathogens described on the WHO priority pathogen list published in 2017 [9]. The prolific scientific literature on plants with significant antimicrobial activity highlights the potential of herbs as credible alternatives in the research and/or development of new drugs against antibiotic-resistant bacteria. The Monsonia plant species have demonstrated an excellent therapeutic potential as a treatment for infectious diseases caused by Gram-negative pathogens such as typhoid fever, dysentery and other ailments such as intestinal hemorrhage [10]. *Monsonia angustofolia* (*M. angustofolia*) found in South Africa has been used by South African indigenous practitioners to treat several health conditions [26]. Scientific investigations on this species have confirmed the presence of phytochemical compounds with pharmacological activity [10]. Another plant of medical interest is *Momordica balsamina Linn* (*M. balsamina Linn*). The plant is reported to have antibacterial, antioxidant and antidiabetic activity [27,28]. The plant methanolic extracts have demonstrated inhibitory activity on the growth of some Gram-negative bacteria such as *Escherichia coli* and *Salmonella typhimurium* [29]. The antibacterial activity of these two plants against clinically relevant Gram-negative bacteria is of great interest. This study investigated their activity against carbapenemase-producing *A. baumannii* in order to identify credible candidates for drug development to address the current antimicrobial crisis and the increasing threat of *A. baumannii*.

## 2. Results

The isolated strain of *A. baumannii* was resistant to all tested antibiotics; carbapenem resistance-associated genes *bla_OXA-51_* and *bla_OXA-23_* were detected. The screening of *M. balsamina Linn* extracts for antibacterial activity using TLC coupled with bioautography did not reveal any level of activity (Table 1). However, using the Kirby Bauer disc diffusion method, only the methanolic extract demonstrated a level of activity against carbapenem-resistant *A. baumannii*. The *M. angustifolia* extracts did not demonstrate any level of activity using the Kirby Bauer disc diffusion (Figure 1). However, using the combined TLC and bioautography method, all the extracts revealed a level of inhibitory action against the clinical isolate of carbapenem-resistant *A. baumannii* (Figure 2 and Table 1). The R_f_ values of antibacterial compounds are presented in Table 2. The methanolic *M. balsamina Linn* extract demonstrated a MIC of 0.5 mg/mL (Figure 3).

## 3. Materials and Methods

### 3.1. Ethical Approval

Ethical clearance to conduct the study was granted by Sefako Makgatho Health Sciences University Research and Ethics Committee under the following reference number: SMUREC/M/219/2020: PG.

### 3.2. Bacteria Collection, Identification and Antimicrobial Susceptibility Testing

The *A. baumannii* isolate was collected at Dr George Mukhari Tertiary Laboratory (DGMTL, Pretoria, South Africa) located at Dr George Mukhari Academic Hospital in April 2018 and identified using a double identification method. The VITEK2 automated system (bioMerieux, Marcy l’Etoile, France) and genotypic confirmation of *A. baumannii* isolate using conventional polymerase chain reaction (cPCR) detection of *bla_OXA-51_* was carried out. Oxacillinase *bla_OXA-51_* is intrinsic to *Acinetobacter species* and reportedly reliable to genotypically confirm strain identity [30]. Antimicrobial susceptibility testing was performed using a VITEK2 automated system (bioMerieux, France). Piperacillin + tazobactam, ceftazidime, cefepime, cefotaxime/ceftriaxone, imipenem, meropenem, gentamycin, ciprofloxacin and trimethoprim/sulfamethoxazole were tested. Results were analyzed and interpreted following the manufacturer guidelines.

### 3.3. Molecular Characterization of Isolate and Identification

The isolate identity as *A. baumannii* was confirmed by a positive cPCR detection of the intrinsic *bla_OXA-51_* and carbapenem resistance, attributed to the presence of *bla_OXA-23_* and *bla_OXA-51_.* Results were analyzed and interpreted in comparison with a known positive control strain run with each experiment. The primers and thermocycling conditions are provided in Appendix A.

### 3.4. Collection of Plant Material

Fresh fruits of *M. balsamina Linn* and *M. angustifolia* whole plant material were collected from Mpumalanga Province in a village known as Phake ya Ratlhagane (25°08′51.1″ S 28°30′28.2″ E) and Ga Mphahlele, Limpopo Province, South Africa (24.305″ S 29.565″ E), respectively, with the assistance of an indigenous health practitioner. Plant specimens were taken to the South African National Biodiversity Institute (SANBI) for botanical identification. A voucher specimen (MA001 and MB001, respectively) was then deposited in the Herbarium of the Pharmaceutical Sciences Unit of the School of Pharmacy, Sefako Makgatho Health Sciences University.

### 3.5. Plant Preparation and Storage

#### 3.5.1. Momordica Balsamina Linn

For preparation of study material, the fruits were cut into halves, left to air dry and the seeds were then taken out. The dried fruits were ground into a fine powder using a Polymix Laboratory Dry Mill Drive Unit (Polymix™ PX-MFC 90 D, Kinematica AG, Luzern, Switzerland) and then stored at room temperature in the laboratory cupboard until further use.

#### 3.5.2. Monsonia Angustofolia

The roots of the plants were cut, and the aerial forms (whole shrub) were rinsed under running tap water. Plant material was left to air dry at room temperature in a well-ventilated room without direct sunlight. Thereafter, dried plant materials were ground into powder using a PolymixTM grinder (PolymixTM PX-MFC 90 D, Kinematica AG, Luzern, Switzerland) and stored in an air-tight container at room temperature for later use.

### 3.6. Preparation of Extracts

#### 3.6.1. Momordica Balsamina Linn

The serial exhaustive extraction method was used to prepare the extracts using four solvents: hexane, dichloromethane (DCM), acetone and methanol, in order of polarity as previously described by Olivier et al., [31]. A quantity of 449.52 g ground plant material with 1.5 L of each solvent was used for the extraction process. The experiment was carried out on an orbital shaker platform (Model 261, 8 kg, 230–50 Hz, 120 W, Labotec, Midrand, South Africa) for 24 h at 50 rpm and repeated three times for each solvent. The resultant supernatant was filtered using filter discs (Filter Discs Qual. 3 hw, 125 mm, BOECO, Hamburg, Germany). The filtrate was concentrated using a Stuart rotary evaporator (RE400, COLE-PARMER LTD. STONE, ST15 OSA, St Noets, United Kingdom) at a temperature of 37 °C with a speed of 50 cycles per minute and the final evaporation of the solvents took place under a stream of air in the laboratory. The dry extracts were weighed, and the extracts were stored at room temperature until further use.

#### 3.6.2. Monsonia Angustofolia

A similar process as described for *M. balsamina Linn* was carried out for *M. angustifolia* with slight modification. Briefly, 500 g of *M. angustifolia* powder was added to 1.3 L of each solvent. The mixture was placed in an orbital shaker for 12 h. The obtained extracts were filtered three times for each solvent and allowed to evaporate at 37 °C using a Stuart rotary evaporator (RE400, COLE-PARMER LTD. STONE, ST15 OSA, St. Noets, United Kingdom). Dried extracts were stored until further use.

### 3.7. Screening for Antibacterial Activity

#### 3.7.1. Disc Diffusion

The Kirby Bauer disc diffusion method was used to screen for antibacterial activity. All tests were carried out in triplicate, standardized to 0.5 McFarland. A freshly recovered colony of *A. baumannii* isolate was lawned onto Mueller Hinton (MH) agar plates (Diagnostic Media Products, Randburg, South Africa). A Whatman No. 1 filter paper was used to make discs, previously sterilized using an autoclave (HL-300 portable, Huxley, Randburg, South Africa). The discs were laced with 5 drops of extract using a sterile pipette and placed on the inoculated agar. The process was carried out for each extract obtained from each plant material using hexane, DCM, acetone and methanol. A known concentration of cefotaxim (30 µg) antibiotic on a disc was used as control for any level of antibacterial activity. The agar plates were then incubated overnight (12–24 h) at 35 ± 2 °C. Finally, the agar plates were assessed for zones of inhibition around the discs.

#### 3.7.2. Thin Layer Chromatography Coupled with Bioautography and Retention Factor Calculation

The standard thin layer chromatography (TLC) coupled with the bioautography method outlined by Famuyide et al., [32], was used with a few modifications to assess the presence of specific compounds within crude plant extract that demonstrate an antibacterial activity. The method integrates the separation and analysis technology of TLC with biological activity detection of bioautography [33]. This is possible with a minimum amount of laboratory equipment and apparatus. The operation is simpler, the experimental cost lower and the sensitivity and specificity higher [33]. This method is suitable for resource-limited settings [33].

The extracts were first redissolved in the solvents that were used for extraction. The preparations were then spotted on the 1.5 mm baseline of a TLC pre-coated silica gel (60 with fluorescent indicator UV_254_). TLC plates were prepared to accommodate one extract at a time. The spotted TLC plates were developed under saturated conditions in their respective mobile phases. The hexane extract was developed with hexane:ethyl acetate (8:2), DCM with hexane:ethyl acetate: acetone (6:2:2), acetone and methanol extracts were developed with hexane:acetone (6:4). The developed plates were set to dry for a period of 10 min to remove traces of the solvent on the plate. The retention factor (Rf) was calculated for each TLC plate. The ratio of distance traveled by a solute to that of a solvent front on a chromatogram system is called the Rf value [34]. This Rf value depends on the concentration and kind of the stationary phase, the concentration and type of mobile phase and the temperature at which the system is run [34].

Thereafter, a fresh overnight-recovered colony of *A. baumannii* was suspended in MH broth and adjusted to 0.5 McFarland. The developed plates were immersed in a bacterial suspension until fully wet. This process was carried out under aseptic conditions. The plates were put to incubation overnight at 35 °C ± 2 with 100% relative humidity. After incubation, the plates were sprayed with a 2 mg/mL solution of p-iodonitrotetrazolium violet (sigma^®^) INT (Fluka Biochemika GA13931, Germany) indicator for visualization of the compound(s) that show activity. A white or creamy band against a purple–pink background was taken as an indication of the presence of compound(s) that have an inhibitory effect against the tested organism [32].

### 3.8. Determination of Minimum Inhibitory Concentration (MIC)

The broth microdilution method was used to determine the MIC of extracts. MIC was determined using the same bacterial strains; 50 µL of MH broth was pipetted into 96-well plates (Thermo scientific, Roskilde, Denmark). Thereafter, 50 µL of the extract solution (1 mg/mL) was added to the first well followed by a two-fold serial dilution. Fifty microliters (50 µL) of an overnight bacterial suspension prepared in MH broth and standardized to 0.5 McFarland was added to each well. The 96-well plate was then sealed and incubated for 24 h at 35 ± 2 °C. After incubation, 2 µL of p-iodonitrotetrazolium chloride (0.2 mg/mL; Fluka Biochemika GA13931, Munich, Germany) was added and incubated for 1 h. The MICs were recorded as the lowest concentrations that inhibited visible bacterial growth.

## 4. Discussion

The oxacillinase-encoding gene *bla_OXA-23_* is the common carbapenamase-encoding gene among *Acinetobacter* spp. The carbapenemase-associated genes *bla_OXA-51_* and *bla_OXA-23_* confer resistance to commercially available antibiotics such as doripenem, ertapenem, imipenem, meropenem. A circulating strain of carbapenemase-producing *A. baumannii* is of concern as highlighted by the WHO [14] and a threat to human life.

When testing *M. balsamina Linn*, the disc diffusion method using the crude methanolic extract showed a level of activity while the combined TLC with bioautography method of the methanolic extract did not have any level of activity (Figure 2 and Table 1). The observed variation in *A. baumannii* susceptibility to the same extracts while using different testing methods might be explained by the nature of the interaction between compounds that formed the crude extract. In disc diffusion, all the compounds are put together and tested against the bacteria, while in the combined TLC with bioautography method the phytochemical constituents of the crude extract are firstly separated and individually tested against the bacteria. Phytochemical molecules are subject to van der Waals interaction [35]. Van der Waals forces are driven by induced electrical interactions between two or more atoms or molecules that are very close to each other and, therefore, influence the behavior of atoms, molecules, structures and compounds [35]. Van der Waals interaction influences a compound’s function. The synergistic and antagonistic effect of the constituents of a crude extract is a plausible explanation for the observed antimicrobial activity of compounds in the extracts [36,37].

The non-polar solvents had more separation of constituent compounds than the polar solvents (Figure 2). Hexane and DCM extracts generally showed better separation of constituent compounds than acetone and methanol extracts. The white inhibition zone on the combined TLC and bioautography plates revealed an antibacterial activity against *A. baumannii* (Figure 2). A previous study has reported the inhibitory action of *M. balsamina Linn* methanolic extract on Gram-negative bacteria such as *Salmonella typhimurium* and *Escherichia coli* [29] that have similar evolution in their resistance pattern to that observed in *A. baumannii* [9]. This additional evidence supports that *Momordica* plant species have a broad-spectrum antibacterial activity against Gram-negative bacteria [29].

The non-polar solvents had a much greater separation of active compounds than the polar solvents (Figure 2; Table 2). The Rf values of active compounds were calculated and are presented in Table 2. There are white inhibition zones on the bioautographs where the purple color did not form due to the presence of active antibacterial compounds against carbapenem-resistant *A. baumannii*. To our knowledge, this is the first report of antibacterial activity with Rf values obtained in these specific conditions using extracts from *M. angustifolia* obtained using the solvent in proportions as described in the methodology.

The recommended clinically significant MIC for a plant extract is 1 mg/mL [38], and as such the methanolic *M. balsamina Linn* crude extract is of clinical interest. Authors have reported that methanolic extract of *M. balsamina* does have better activity on Gram-positive bacteria than Gram-negative bacteria [27]. A significant MIC of 0.4 mg/mL was observed while testing the methanolic extract of *M. balsamina* against methicillin-resistant *Staphylococcus aureus* (MRSA) [27]. MRSA is also listed as one of the top priority pathogen as the *A. baumannii* strain in this study [9]. The highest concentration of 1 mg/mL considered as the starting concentration for MIC determination did not enable us to identify a clinically relevant MIC with the other extracts. No crude extract from *M. angustifolia* showed activity against carbapenemase-producing *A. baumannii*, both by the Kirby Bauer disc diffusion method and the MIC determination. However, Mcotshana et al., [39] in March 2022 reported an antimicrobial activity from *M. angustifolia* against fungal microorganisms such as *Cryptococcus neoformans* and *Candida albicans* [39] with extracts of DCM and methanol with concentrations lower than 1 mg/mL. Thus, these researchers highlighted that the same extract did have a level of antibacterial activity against Gram-negative bacteria such as *Escherichia coli* (ATCC 25922) and *Pseudomonas aeruginosa* (ATCC 27853) and Gram-positive bacteria such as *Staphylococcus aureus* (ATCC 29213) if concentrations above 1 mg/mL were used [39]. These results combined with this current report pinpoint the potential of *M. angustifolia* as an antifungal agent rather than antibacterial agent.

## 5. Conclusions

Local extracts of *M. angustifolia* and *M. balsamina Linn* using various solvents have revealed a level of antibacterial activity against a carbapenemase-producing *A. baumannii* clinical isolate. South African *M. balsamina Linn* methanolic extracts have demonstrated clinically relevant antibacterial activity against a carbapenemase-producing *A. baumannii* isolate. Although additional investigation to elucidate the active compound is required, *M. basalmina Linn* stands as a credible candidate for antibacterial drug development.

## Figures and Tables

**Figure 1 plants-11-02374-f001:**
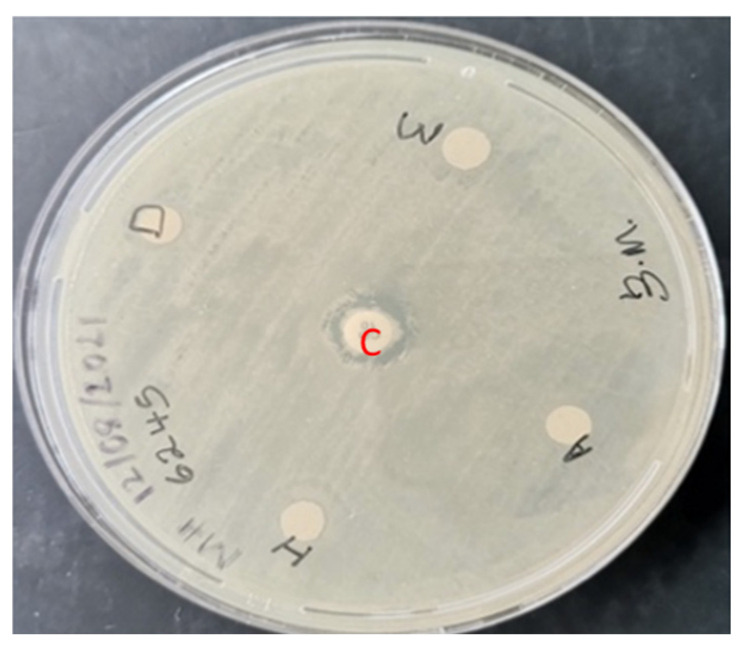
Screening of *M. angustifolia* plant extract obtained using four different solvents against *A. baumannii* isolate using Kirby Bauer disc diffusion. A: Acetone solvent; H: Hexane solvent; D: Dichloromethane solvent; M: Methanol solvent; C: Cefotaxim (30 µg) antibiotic used as control.

**Figure 2 plants-11-02374-f002:**
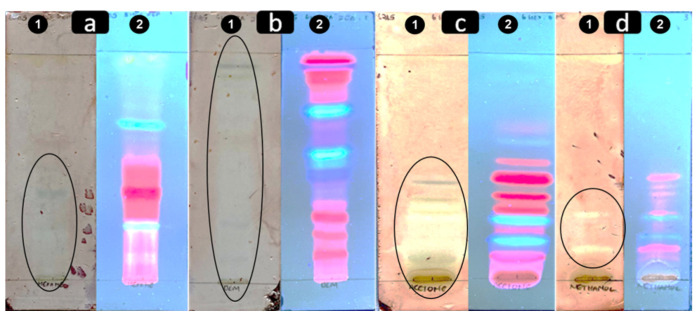
Developed TLC–bioautograph of *M. angustifolia* extracts representing compounds that have antimicrobial activity against *A. baumannii*. (**a1**) Combined TLC and bioautography screening method for hexane extract; (**a2**) developed hexane extract TLC plate visualized under UV; (**b1**) combined TLC and bioautography screening method for DCM extract; (**b2**) developed DCM extract TLC plate visualized under UV; (**c1**) combined TLC and bioautography screening method for acetone extract; (**c2**) developed acetone extract TLC plate visualized under UV; (**d1**) combined TLC and bioautography screening method for methanol extract; (**d2**) developed methanol extract TLC plate visualized under UV. The elliptical circles show the clear zone where bacterial growth has been inhibited.

**Figure 3 plants-11-02374-f003:**
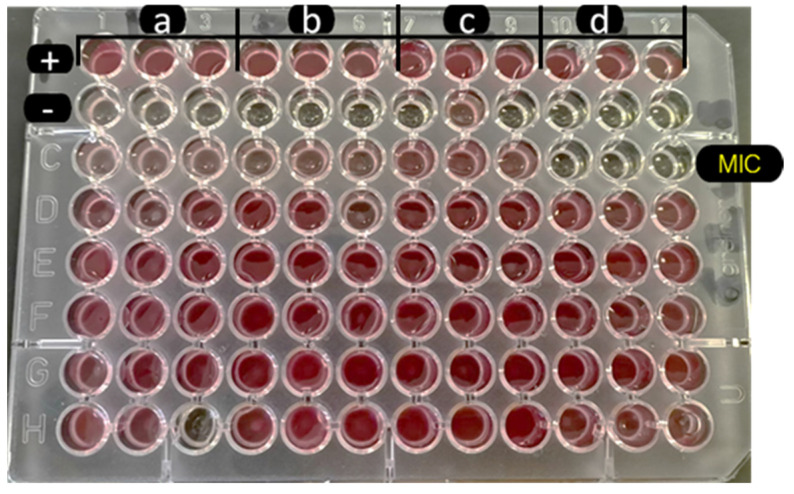
MIC determination of methanol *M. balsamina Linn* extract that inhibits *A. baumannii* growth. +: Positive control; -: Negative control; (**a**) hexane extract; (**b**) DCM extract; (**c**) acetone extract; (**d**) methanol extract; MIC: Determined MIC of methanol extract. Each experiment was carried out in triplicate.

**Table 1 plants-11-02374-t001:** Antibacterial activities of plant extract against *A. baumannii*.

Plants	Testing Method	Solvents			
		Hexane	DCM	Acetone	Methanol
*Momordica balsamina L*	K.B.D.D. TLC/Bioautography	- -	- -	- -	+ -
*Monsonia angustofolia*	K.B.D.D. TLC/Bioautography	- +	- +	- +	- +

K.B.D.D.: Kirby Bauer disc diffusion; -: Antibacterial activity not detected; +: Antibacterial activity detected.

**Table 2 plants-11-02374-t002:** R_f_ values of active compounds from *M. angustifolia* representing *A. baumannii* growth inhibition.

Scheme	Hexane	Dichloromethane	Acetone	Methanol
R_f_ Value	0.05 0.17 0.39 - - - - - -	0.58 0.78 0.36 0.48 0.5 0.56 0.67 0.78 0.9	0.11 0.56 0.24 0.37 - - - - -	0.11 0.27 - - - - - - -

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
