# Peer review of "Antibacterial Activities of Monsonia Angustifolia and Momordica Balsamina Linn Extracts against Carbapenem-Resistant Acinetobacter Baumannii"

_plants, 2022, doi:10.3390/plants11182374_

Round 1
Reviewer 1 Report
The results of the activity against Actinobacter boumani, in comparison with the activity against bacteria, previously published for both species is not discussed. No mention is made that the molecules identified in both species have antimicrobial activity. From Monsonia angustifolium seven compounds were isolated and tested as antimicrobials.
See: (not included in the references of the manuscript) Zenande K.S., et al., (2022) "Cytotoxicity and antimicrobial activity of isolated compounds from Monsonia angustifolia and Dodonea angustifolia" J. Ethnopharm., on line, 28 March, 2022, 115170

Reviewer 2 Report
The introduction is well written and provides enough data to understand the topic and the main objectives to be achieved.
Error line 47: ventilator-associate54d pneumonia
Materials and methods:
I would have liked to see photographs of the making of the Thin Layer Chromatography coupled with Bio-autography. At least in the supplementary material
Results:
Figure 1 is shown first, but table 1 is commented out first, change the order
I don't see what the use of the Thin Layer Chromatography coupled with Bio-autography method brings to the paper. What relevant data do the authors obtain from them in Fig 1?
Figure 1: Which aims to indicate the 4 elliptical circles in the figure. Explain it in the legend
Table 1: I would like to see the photos of the different plates, or at least the most representative
Table 2: It is not clear to me how these data were obtained or what they represent? What method was used? what units are? It is necessary to show the standard deviations.
Figure 2 legend: the concentrations used in each of the wells are not indicated. I don't see it correct to show the results that way, with a simple photograph. Why haven't the authors determined the optical density of each well in each of the experiments, calculated the mean deviation, and presented the data in graphical form? I recommend that it be done
In the Line 208-210 the authors says: However, using the Kirby Bauer disc diffusion method only the methanolic extract demonstrated a level of activity against carbapenem-resistant A. baumannii. But I see that, this statement does not agree with the data in table 1, it says the opposite.
Discussion
The discussion is well written, but it does not contribute much more than what is seen in the results, I would recommend in this short and concrete paper to do the results and the discussion together.
Reviewer 3 Report
I read with interest this short study, which reports the initial antimicrobial screening of different plant extracts. One of the techniques adopted, bioautography, is not extremely accurate, but it seems suitable for a preliminary characterization. The use of actual clinical strains producing carbapenemase is definitely commendable. Overall, the manuscript is OK, and can be published after a minor revision. The revision is mainly needed to make the manuscript more readable and accessible to non-specialist, like researchers from neighbouring areas.
1. In the introduction, explain briefly what Bio-autography is and why it was preferred over more modern and informative techniques for antimicrobial characterization of plant extracts.
2. Explain in the methodology rather than at the end of the discussion section, what Rf is.
3. In other works on antimicrobial characterization of plant extracts, the TLC bands corresponding to compounds with significant antimicrobial activity were scraped, redissolved and tested in microdilution broth assay. Did you do such experiment to resolve the antimicrobial activity of the plant extract?
4. Did you attempt a preliminary characterization of the chemical compounds in the antimicrobial TLC bands? This will significantly add to the novelty adn impact of the work reported.
Minor comments
Line 163. One or more words are missing. Please check.
Line 185. One of more words are missing. Please check.
Round 2
Reviewer 2 Report
Dear authors,
I consider that the authors have responded correctly to most of my suggestions, and therefore I consider the paper can be accepted for publication.